# Harnessing long-term gridded rainfall data and microtopographic insights to characterise risk from surface water flooding

**Kriti Mukherjee** *, **Mónica Rivas Casado, Rakhee Ramachandran, Paul Leinster**

School of Water, Energy and Environment, Cranfield University, Bedford, United Kingdom

* kriti.mukherjee@cranfield.ac.uk

## Abstract

Climate projections like UKCP18 predict that the UK will move towards a wetter and warmer climate with a consequent increased risk from surface water flooding (SWF). SWF is typically caused by localized convective rainfall, which is difficult to predict and requires high spatial and temporal resolution observations. The likelihood of SWF is also affected by the microtopographic configuration near buildings and the presence of resilience and resistance measures. To date, most research on SWF has focused on modelling and prediction, but these models have been limited to 2 m resolution for England to avoid excessive computational burdens. The lead time for predicting convective rainfall responsible for SWF can be as little as 30 minutes for a 1 km x 1 km part of the storm. Therefore, it is useful to identify the locations most vulnerable to SWF based on past rainfall data and microtopography to provide better risk management measures for properties. In this study, we present a framework that uses long-term gridded rainfall data to quantify SWF hazard at the 1 km x 1 km pixel level, thereby identifying localized areas vulnerable to SWF. We also use high-resolution photographic (10 cm) and LiDAR (25 cm) DEMs, as well as a property flood resistance and resilience (PFR) database, to quantify SWF exposure at property level. By adopting this methodology, locations and properties vulnerable to SWF can be identified, and appropriate SWF management strategies can be developed, such as installing PFR features for the properties at highest risk from SWF.

## 1. Introduction

With climate change and population growth, a wide range of homeowners living in highly populated areas are increasingly vulnerable to flood or drought events. Around 5.6 million properties in England are currently at risk from surface water flooding (SWF) [1]. Of these, currently around 325,000 properties are in areas at the highest risk–meaning there is a more than 60% chance they will flood in the next 30 years. Without action, up to 295,000 more properties could be put at risk by 2055 [2]. By 2050, 90% of the UK population is projected to live within urban areas [3] and the damages from floods have been projected to increase in the UK by 60–220% in the next five decades [4]. Governmental agencies across the world have

**Data Availability Statement:** "*****A/PROS AT ACCEPT: Please follow up with the authors to make

**Funding:** We acknowledge EPSRC funding EP/N010329/1 for Building Resilience into Risk Management (BRIM).

**Competing interests:** The authors have declared that no competing interests exist.

developed policy strategies, such as the Flood Risk Directive (2007/60/EC; [5]) published by the European Union, to protect homeowners from increased SWF and fluvial flooding. In England, the highest recorded level of damage occurred in the 2007 flood incidents, when 55,000 properties were flooded, and 40% of the flood damage was related to pluvial or SWF, for which no forecasts, models or management strategies existed [6,7]. Such incidents and the Flood Risk Directive prompted the government via the Environment Agency to generate flood risk maps for England [7]. Based on the risk estimates, SWF was included in the UK national risk register for the first time in 2016 [8,9]. Research resources dedicated to understand and manage SWF have been hitherto much less compared to fluvial flooding [10,11].

SWF events are typically caused by short-term intense rainfall, which can overwhelm urban drainage systems and lead to flooding. These types of rainfall events, also known as convective rainfall, occur when warm, less dense air rises and condenses, leading to precipitation. Because convective rainfall is localized and difficult to predict, it poses a significant challenge for SWF management. This is especially true in urban heat islands, where temperatures can be significantly higher than in surrounding areas, leading to more frequent and intense convective rainfall events. There is ongoing research into improving the prediction and management of SWF events, with a particular focus on the use of high-resolution rainfall data and advanced modelling techniques [12,13].

Another challenge to managing SWF risks is a lack of knowledge of the microtopography of areas affected by SWF [14]. Microtopography (e.g. heights of the buildings, fences, walls, speed bumps, kerbs, channels, roads) plays an important role in altering the hydrological, hydrodynamic and hydromorphological SWF processes, which in turn affects SWF depth, velocity and extent estimations [15,16]. Microtopography based on sub-meter level high resolution DEMs provides information about the natural processes occurring in a catchment [17]. Several primary and secondary attributes based on DEMs can be used to extract additional information about the landscape and corresponding hydrological conditions [18]. The most used secondary attribute, based on a combination of primary attributes such as slope and elevation, is the topographic wetness index (TWI), which describes the tendency of each grid of a DEM to accumulate water due to the local topographical effects [17,19]. More about TWI is discussed in the following sections. Microtopography can also include information about property resilience and resistance measures (PFR) [20] (e.g., flood aperture guards for doors and windows, flood resistant airbricks, and raised doors or steps leading to a property). Two buildings at similar elevations and slopes may be affected by the same depth of SWF differently depending on the presence or absence of PFR. These features restrict the flow of water into the building and thus influence the impact of SWF on a property [21].

Uncertainties in long term flood prediction due to underlying climate trends is another challenge to managing future SWF [22]. The uncertainties in the projected future rainfall need to be considered in assessing the uncertainties in modelled flood depth and associated flood risk estimates.

Following the definition by Defra and Environment Agency [6], flood risk is comprised of flood hazard, exposure, and vulnerability. Hazard is characterized by extreme rainfall events resulting in water accumulation, flow, and debris. Exposure consists of flood warning, speed of onset of flood, and characteristics of buildings depending on the number of floors and the presence or absence of flood resistance features. Vulnerability characterizes the risk of individuals to flood events depending on their age and health.

The aim of this study is to develop a conceptual framework based on microtopographic information to quantify location specific SWF hazard and exposure at property level. This will be achieved through three specific objectives as follows:

O1. To characterize the localized rainfall SWF hazard based on the frequency of extreme wet months, number of daily extreme rainfall events, and maximum daily rainfall amount.

O3. To characterize SWF exposure at property level based on microtopography and PFR data.

## 1.1 Study area

We have studied SWF hazard and exposure in the market town of Cockermouth (Northwest Cumbria, UK), which is located at the confluence of the Cocker and Derwent rivers, that makes it susceptible to flooding (Fig 1). Cockermouth has an area of 2.8 km$^2$, and a population of 7,900 in 2016 [23] that increased to approximately 9,500 as of 2021 [24] (Fig 1). Cockermouth is quite vulnerable to flood incidents with 15 flood events recorded since 1751 [23]. Four of these happened in the period after 2000 (2005, 2008, 2009, 2015) [23]. This implies that the frequency of flood incidents increased considerably in the recent decades. In December 2015, following Storm Desmond, a record rainfall of 405 mm in 48 hours was measured that damaged 466 properties [23].

## 2. Data collation and collection

We collated gridded precipitation data for Cockermouth available at 1 km spatial resolution to calculate the localized SWF hazard. To validate the gridded modelled precipitation data, we

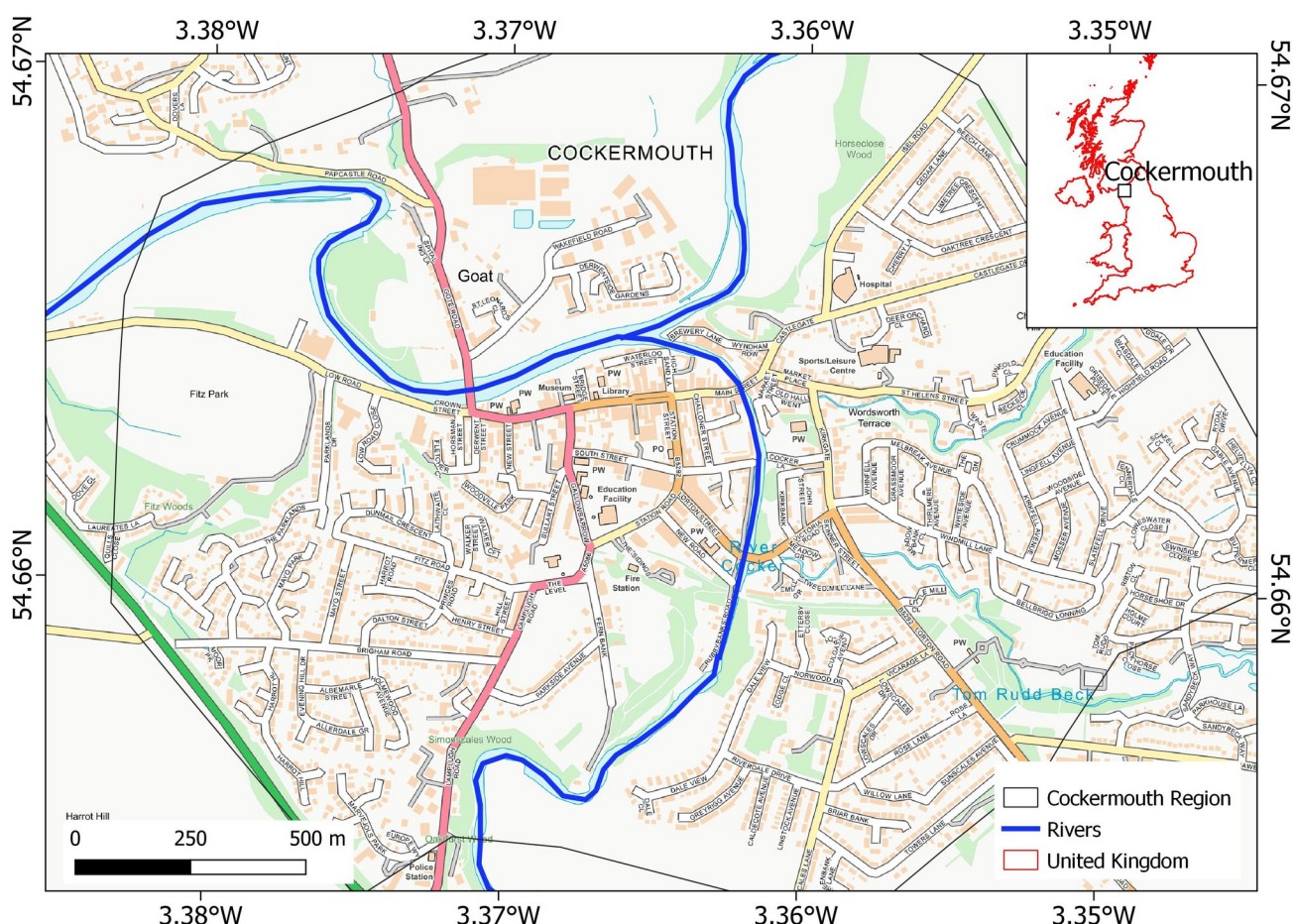

**Fig 1. Study area: Cockermouth town.** The Basemap is available from Ordnance Survey (OS OpenMap–Local) and contains public sector information licensed under the Open Government Licence v3.0.

used remotely sensed precipitation observation from NASA's Global Precipitation Measurement (GPM) data. In addition, to assess SWF exposure we collected field survey data to identify PFR measures in Cockermouth. For microtopography information we used DEMs at 25 cm and 10 cm spatial resolutions. Below we describe each of these data sets in more detail.

## 2.1. Gridded precipitation data

The HadUK data (1 km spatial resolution) is produced by the UK Met Office Hadley Centre for Climate Science and Services. HadUK provides gridded climatic observations for the UK by combining multiple regressions with inverse distance weighted interpolation of irregular in situ observations to regular grids [25]. Factors such as latitude, longitude, altitude, topography, coastal influence and urban effects are taken into account while performing the interpolations [26]. The daily total rainfall data is available from 1891. In this study, we use a subset of the daily total rainfall data—from 1951 to 2020. We compared the data from 1951 to 1985 with that from 1986 to 2020 to see if there was an increase in flood hazard from extreme rainfall events in the more recent period.

## 2.2. Precipitation validation data

HadUK data has gone through rigorous validation and quality check before being available in the public domain [25]. However, as there are no historical station observations near Cockermouth, Cumbria, we used a satellite based remotely sensed precipitation product known as Integrated Multi satellitE Retrievals for GPM (IMERG) [27]. The precipitation estimates are obtained by integrating satellite microwave and infrared precipitation estimates with precipitation gauges and other precipitation estimators for the entire globe. Therefore, this precipitation estimates are expected to closely represent the actual ground conditions [27,28]. This data is available at 10 km spatial resolution. We used the research level GPM product that uses the data from Tropical Rainfall Measuring Mission (TRMM: available from June 2000 to April 2015) [29] and the GPM (available from April 2015 to September 2021) eras, thus covering June 2000 to September 2021 in total.

## 2.3. Microtopography data

Elevation information was collected using stereo photos captured by a photographic camera mounted on an Unmanned Aerial Vehicle (UAV) and laser pulses using a laser scanner mounted on an aircraft.

**2.3.1 Stereo air photo DEM.** We collected stereo air photos in the visible region of the electromagnetic spectrum. We used a Sirius Pro Unmanned Aerial Vehicle (UAV) and a Sony Alpha ILCE 6300 PRO camera (Sony Europe Limited, Weybridge, Surrey, UK) to capture the photos. The take-off payload of the platform was 2.7 kg and the flying height was 125 m. The UAV was powered by a 5300 mAh battery, had a wingspan of 163 cm, and a length of 120 cm and included a Global Positioning System (GPS: GNSS-RTK- L1/L2 and GLONASS with a planimetric accuracy of 0.01 cm and altimetric accuracy of 0.015 cm) and an Inertial Measurement Unit (IMU) to measure camera positions and orientations (yaw, pitch, and roll). An overlap of 65% -85% was maintained for consecutive photographs. The survey was conducted by qualified UAV pilots between 17th and 19th July 2019 (cloud free and excellent visibility) and the DEM was generated at 10 cm spatial resolution with a vertical and horizontal positional accuracy better than ±10 cm [1].

**2.3.2 Light Detection and Ranging (LiDAR) DEM.** We collected LiDAR Data using a Teledyne Optech Galaxy (Teledyne Optech, Toronto, Canada) topographic laser from a Partenavia P68 aircraft. The laser scanner emitted radiation at 18 pulses per m$^2$ (ppm) to generate

high-density topographic data. The flight plan included ten overlapping swaths with an ancillary perpendicular swath across the surveyed area. The Optech Galaxy had the PulseTRAKTM technology to provide high pulse repetition frequencies (PRFs) at high altitudes and Swath-TRAKTM technology to achieve a constant swath width on the ground by dynamically changing the scan Field-of-View (FOV). The camera coordinates and orientation information were recorded by GPS (Trimble Applanix L1/L2 Card) and IMU. The data was collected on 23$^{rd}$ July 2017 in clear weather conditions from a flying height of 600 m. The DEM was generated at 25 cm resolution with a positional accuracy better than ±10 cm [1].

## 2.4. Property flood resilience and resistance (PFR) data and building database

PFR data for a total of 365 properties in Cockermouth were collected in 2017. These properties included all households considered to be affected by the 2015 flood event caused by Storm Desmond [30]. The additional flood depth protection provided by the PFR measures was measured at cm level. Data were collected via a Samsung SM T560 8GB tablet (Samsung 155 Electronics Co. Ltd., Seoul, Korea) using the ESRI Collector Application (ESRI, Redlands, 156 CA, USA) for Android. The height of the flood protection was added as an attribute to the OS (Ordnanace Survey) Master Map geospatial vector data available for free [31].

## 3. Methods

We estimated the SWF hazard based on rainfall data. This is a variable quantity that depends on the period of observation and resolution of the climate data. We estimated the local SWF hazard at pixel level to visualise whether the flood hazard has increased over time. We estimated SWF exposure at property level using the high-resolution LiDAR and photographic DEMs and the PFR database.

## 3.1 Rainfall surface water flood hazard estimation

Within the context of SWF hazards, we focused on the characterization of localized rainfall. Accurate localized rainfall observations and forecasts at high spatial and temporal scales are recommended to improve the performance of meteorological (rainfall) and hydrological (flood) models and better inform SWF management decisions [13,32]. If SWF warnings are issued based on observations from rain gauge stations, then a density of one gauge per km$^2$ is usually necessary [33]. Even at this resolution, the lead time for SWF forecast is only 30 minutes for a 1 km x 1 km storm [13]. The density of rainfall gauging stations in the UK is high (7 x 7 km$^2$) [26] but does not reach this benchmark, which makes accurate SWF forecasting still quite challenging. The existing network also varies with time due to changes in the number and/or location of stations, leading to discontinuous measurements in time and space [26] and inaccuracies in confidence, precision, and lead times for the SWF forecasts [34]. Extreme rainfall events are comparatively rare, and continuous long rainfall records are required to differentiate between normal and extreme rainfall events both spatially and temporally [35]. Such information is not available for the whole of England based on the available rain gauge/weather stations with irregular spatial and temporal observations. In this study, we sought to understand whether, in the absence of long-term spatially and temporally continuous observation network, modelled or interpolated long term gridded rainfall data could be used to represent the flood hazards. To validate the data, we compared our results with some of the historical observations of flood events in Cockermouth.

   We used the monthly and daily total rainfall data to identify the extremely wet months and days. Also, the maximum daily rainfall helped us understand how this has increased over the

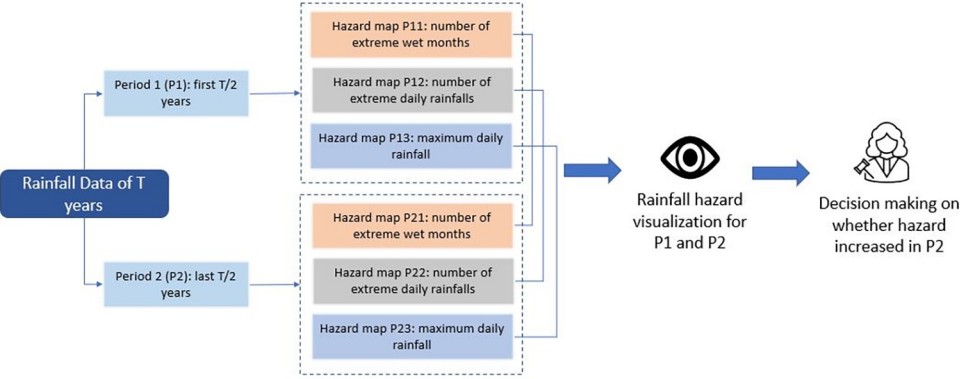

**Fig 2. Framework for SWF hazard estimation.**

period. As the number of extreme wet periods, extreme rainfall days or maximum daily rainfall increases, SWF hazard also increases. In our framework (Fig 2), we divide the data in two different equal periods, period 1 representing the first half of the data, and period 2 representing the second half of the data, to understand whether the hazard from SWF has increased in the period 2 compared to period 1.

The Standardised Precipitation Index (SPI) is used to understand whether a certain period is wetter or drier than normal. This index was originally proposed by [35]. SPI ranges between -∞ and +∞. Values < -2 imply extreme dry conditions whereas values > 2 imply extreme wet conditions [35]. This index is valid for monthly or longer extreme rainfall events. We use this index at a monthly scale to identify the extreme wet periods that could lead to fluvial and/or SWF events [36–38]. We used monthly precipitation values from HadUK data to understand how these have changed over time. We used the data from 1951–1985 (period 1) to generate an average climate for each grid. We fitted a gamma probability distribution function, as proposed by [39,40] to the monthly frequency distribution of precipitation for period 1 as described by Eq 1.

$$f(y) = \frac{y^{\alpha-1} e^{-\frac{y}{\beta}}}{\beta^{\alpha} \Gamma(\alpha)} \tag{1}$$

where, y is the monthly total precipitation, α is the shape parameter and β is the scale parameter of this distribution. We use these parameters to find the cumulative probabilities of the observed precipitations from both period 1 and period 2. The next step in the SPI calculation, as described in [35], is to transform these values to the corresponding standard normal values with mean zero and standard deviation one. This is implemented for each month and each grid, so that a zero value corresponds to no deviation from normal precipitation; positive values indicate more than normal precipitation; and negative values indicate less than normal precipitation. As described by [35], we consider all values >2 as extreme rainfall events and tag each of them as Flag = 1. Then we sum up the number of flags for period 1 and period 2 to find the total number of extreme wet months in each period (Fig 2).

We use the daily rainfall values to identify extreme rainfall days. The daily rainfall values do not follow a gamma distribution and we standardise the daily rainfall series (x) to a new series (x′) by the following mapping [41]:

$$x'_i = \frac{(x_i - \mu)}{\sigma} \tag{2}$$

Where, $x_i$ is the original value on the i'th day of each period, μ is the mean of all the original rainfall values in period 1, σ is the standard deviation of the rainfall values in period 1 and $x'_i$ is the transformed rainfall on the i'th day. By this standardisation, we can represent the values from different locations and different times on a common scale so that they are comparable. We use the 99 percentile of all the standardised daily rainfall values in period 1 as the threshold above which all daily rainfall events are considered as extreme rainfall events in both periods. We then count the number of such events in both periods to generate the hazard maps (Fig 2).

Finally. we used daily total rainfall data from each period to extract the maximum daily rainfall for each period. To visualise the change in SWF hazard, we represent each of the three types of hazards for the two different periods of each pixel on a common scale for both period 1 and period 2.

**3.1.1. Representativeness of HadUK data for Cockermouth Region.** HadUK data is generated from pre-processed station observations extracted from the Met Office's quality controlled Integrated Data Archiving System (MIDAS, [42]). The data was then validated using collocated station observations to see the agreement and determine thresholds to remove suspect data as outliers. Less than 0.1% of the daily data and between 0.1% and 0.4% of the monthly data were excluded as outliers [25]. However, as there is no meteorological station in Cockermouth Region, we validated HadUK data for this region with GPM observations. We compared the monthly precipitation values from June 2006 to September 2021 (availability of GPM data and overlapping period of the two data sets) and computed cross-correlation at different lags. Pearson's coefficient for each cross correlation were calculated to identify the most significant (p value < 0.05) relationship with maximum correlation. There were two pixels at 10 km resolution covering the study area. We reported on the extreme rainfall events in Cockermouth in the recent period (1986–2020) using these two pixels and compared these with reported flood incidents.

In addition, we focused in detail on the Storm Desmond flood event (December 2015) in Cockermouth to understand whether the daily precipitation of HadUK data provides a robust estimate for this incident. The Environment Agency [23] reported prolonged rainfall events from 4th to 7th December 2015, with more than 300 mm rainfall in 24 hours. We, therefore, plotted the rainfall from 1st to 10th of December 2015 using HadUK and GPM data and visually checked the closeness of these representations.

## 3.2 Surface water flood exposure estimation

Within the context of SWF exposure, we focused on two characteristics of households: the configuration of the microtopography around the property and the presence of PFR measures. The resolution of the DEMs used for flood modelling influences the accuracies of modelled flood extents/depths, and the accuracy of flood modelling increases with improvement in the underlying spatial resolution [15,16,43]. Current flood modelling approaches enable the estimation of flood extent and impact at post-code level [44]. There is always a trade-off between resolution and computing efficiency, and the use of high spatial resolution data (better than 2 m) in flood models results in a high computational burden [43]. Here, we intend to characterize SWF property level exposure using a metric based on microtopographic information only, thus avoiding the use of flood models. We used submeter (10 cm and 25 cm) level resolution DEMs for Cockermouth. Webber et al. [45] reports that PFR measures can protect households to up to 0.6 m of flood depth, thus reducing the damage from surface water to a great extent. Unfortunately, the inclusion of data detailing household level PFR measures presence is limited in flood models. This is primarily due to a lack of a database with such information [21]. In this study we built a property level flood resilient and resistant measures database for

Cockermouth and used this information to further refine the microtopographic characterization. Flood exposure is therefore estimated at property level using high resolution elevation and PFR vector data.

**3.2.1 Microtopographic influence on SWF exposure.** Topography plays an important role in controlling the spatial variation of hydrological conditions and can provide information about the natural processes occurring in a catchment [17]. Digital Elevation Models (DEMs) provide information about the topography of a landscape. Several primary and secondary attributes based on DEMs can be used to extract additional information about the landscape and corresponding hydrological conditions [18]. The most used secondary attribute, using a combination of primary attributes such as slope and elevation, is the topographic wetness index (TWI) that describes the tendency of each grid of a DEM to accumulate water due to the local topographical effects [17,19]. This is based on a concept to identify wet areas that are highly exposed to overland flash flooding and is defined by Eq (3).

$$TWI = \ln\left(\frac{A}{\tan(\varphi)}\right) \tag{3}$$

where A is the upslope catchment area, draining across a unit width of contour [17], and $\varphi$ is the steepest downslope direction. We assume the properties of soil to be uniform, i.e. the transmissivity is constant throughout the landscape and has a value of unity. Areas prone to water accumulation (e.g. large drainage area contributing to accumulation) and low slope angle have high TWI values, whereas well drained dry areas have low TWI values [19]. *A* determines the area contributing to the draining of water into a pixel. A flow routing algorithm is used to determine *A*, which establishes the direction of the flow [46].

First, we excluded the vegetation and water pixels from the DEMs by visual interpretation from the true colour image captured by the UAV. Then we calculated TWI within a geographic information system environment using QGIS [47] PCRaster Tools plugin as follows:

$$TWI = \frac{flowAccumulation \times pixelArea}{\tan(SR)} \tag{4}$$

where, SR is the slope of each pixel in radians; *flowAccumulation* is the accumulated amount of water in each pixel obtained from a local drain direction map that determines the direction for water flow from each pixel to its steepest downslope neighbour (among eight surrounding pixels); pixelArea is the area of each pixel of the DEM. Pixels with lower values of TWI are usually regions having steep slopes or crests and subsequently have a high tangent value. Pixels with higher TWI values indicate higher vulnerability to water accumulation. These are the regions having low slopes ($\phi$) and a large upslope contributing area (*A*).

We calculated the effect of microtopography on a property's flood exposure by first creating a 15 m wide buffer polygon for each building. For demonstration purposes, we assumed that 15 m was sufficient to capture the exposure from SWF for each building. We then extracted the TWI raster image for each of the polygons. Based on the distributions and to avoid the influence of outliers, we assumed that the median of the raster values in each polygon represented the exposure of each building to SWF due to water accumulation in the neighborhood and based on microtopography. We carried out this exercise for the TWI image generated using both DEMs (UAV-RGB and aircraft-LiDAR) to assess the sensitivity of the framework to different types of microtopographic input information. For that purpose, we built a confusion matrix between the risk categories of different buildings identified by the two different DEMs.

**Table 1. Flood exposure category corresponding to exposure metric.** EM: Exposure from microtopography; EH: Exposure from PFR height.

| Exposure class | Exposure category | EM | EH |
|---|---|---|---|
| Very low | 1 | Min$\leq$EM$<$25 | EH$\geq$1 |
| Low | 2 | 25P$\leq$EM$<$50 | 0.5$<$EH$<$1 |
| Medium | 3 | 50P$\leq$EM$<$75P | 0$<$EH$\leq$0.5 |
| High | 4 | EM$\geq$75P | EH = 0 |

**3.2.2 Calculating SWF exposure from PFR data.** Using the PFR database and the OS Mastermap building data for Cockermouth, we refined the SWF exposure of each building. First, we generated an exposure metric based on the height of the flood resistance feature present in the building. We assume any protection more than or equal to 100 cm height to reduce the SWF exposure to zero. This assumption is based on the observations by [14] that the depth of water due to SWF in Cockermouth typically does not exceed 100 cm. We assumed properties with no PFR protection (equivalent to 0 cm height) to have a maximum exposure (1). To the flood protection heights (h) within 0 and 100 cm, we fitted a modified sigmoid function (Eq 5).

$$Flood \exp o\ sure = \frac{2}{1 + e^{0.03*height}} \tag{5}$$

Thus, using the PFR height data and Eq (5), we calculate the SWF exposure at property level from PFR (EH$_i$, i = 1,2,...,N, here N = 365).

**3.2.3 Classification of buildings according to flood exposure.** We calculated the 25, 50, and 75 percentiles of the distribution defined by the exposures from microtopography (EM). We used the calculated percentile values as thresholds to categorize the exposure in four classes as shown in Table 1.

For the PFR based exposure (EH), we categorise metric 1 as lowest risk and of exposure class "Very Low", and metric 0 (corresponding to no PFR) as category 4 at the highest risk with exposure class "High". The intermediate values are categorised in exposure categories Low and Medium as shown in Table 1.

Then, using the exposure category for each building arising from EM and EH, we find the resulting exposure categories by using Fig 3. Each element in Fig 3A is the sum of exposure categories in the corresponding row and column. We assume that when the exposure from one of the two categories is very low, the overall exposure is very low. This indicates a sum of

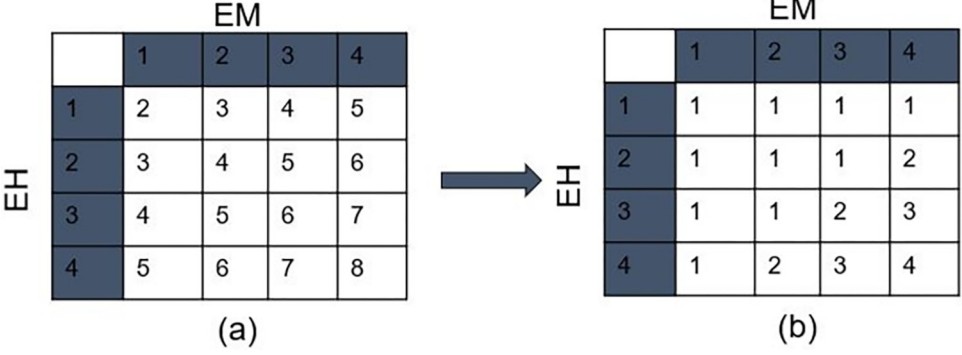

**Fig 3. Property flood exposure as a result of exposures from microtopography and PFR.** (a) Summation of exposures from corresponding rows and columns, (b) Final property flood exposure categories.

maximum 5 using both exposures and accordingly we use all elements having values ≤5 in this category to have a resulting exposure of very low in Fig 3B. All elements having a summation of 6, 7, and 8 are classified as having exposures low, medium, and high, respectively (Fig 3).

## 4. Results

### 4.1 Localised surface water flood hazard from HadUK data

For Cockermouth, the total number of extreme wet months (SPI>2) for different pixels using HadUK data for the two periods studied are shown in Fig 4. The minimum and maximum numbers for extreme wet months for period 1 are 3 and 7 respectively, and those for period 2 are 8 and 17 respectively. This clearly indicates an increase in the number of extreme wet months in the recent period compared to the earlier period.

There are within 127 and 144 days of extreme rainfall events in period 1 and within 142 and 189 extreme rainfall days in period 2 (Fig 5). This again implies an increase in SWF hazard in the recent period compared to the earlier period.

The maximum daily rainfall levels were within the range 53 mm and 85 mm for different pixels (total 30 pixels, Fig 6) in period 1. The corresponding values in period 2 were between 66 mm and 91 mm (Fig 6).

Based on HadUK data, it appears that the monthly rainfall has considerably increased in Cockermouth (Fig 4, within the black polygon representing Cockermouth town), more than the surrounding regions. whereas there is a medium increase in daily extreme rainfall and maximum daily rainfall (Figs 5 and 6) in this region in comparison to the surrounding regions.

**4.1.1 Representativeness of HadUK data for Cockermouth Region.** We analysed the extreme rainfall events separately for the nearest two pixels to the rivers Cocker and Derwent based on HadUK data. The values indicate that a monthly rainfall ≥200 mm has been used by

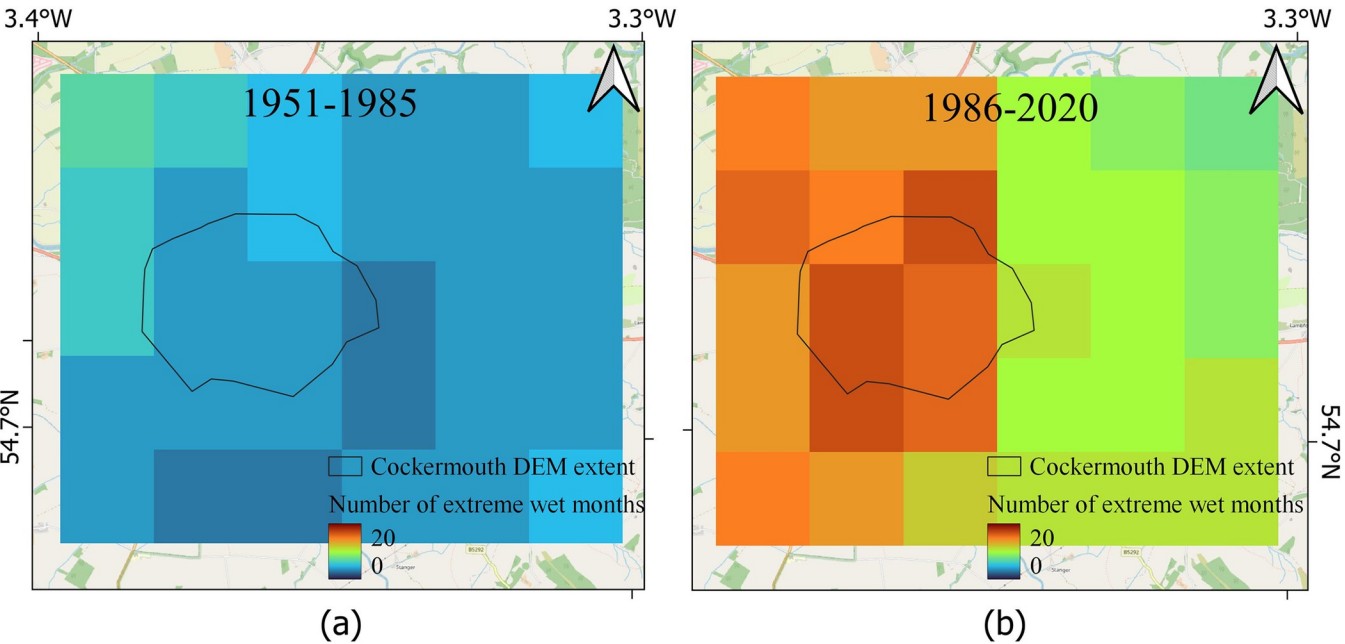

**Fig 4. Estimates of flood hazards for Cockermouth from extreme wet months.** (a) The number of extreme wet months in 1951–1985, (b) in 1986–2020.

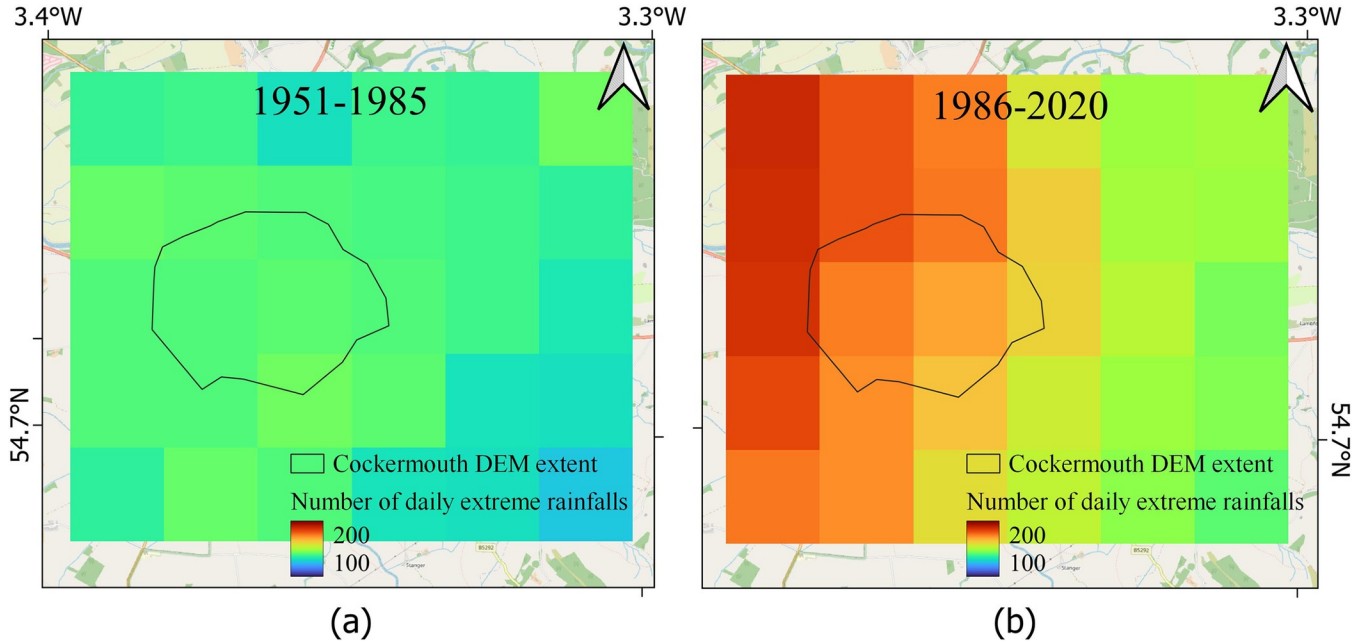

**Fig 5. Estimates of flood hazards for Cockermouth from extreme daily rainfalls.** (a) The number of extreme daily rainfalls in 1951–1985, (b) in 1986–2020.

the gamma distribution as extreme for these two grids. A total of 16 events, of which 4 are between 1986 and 2000 and the rest are after 2000, were identified (Fig 7). This clearly indicates an increase in wet periods and related flood hazards in Cockermouth in the recent period. The months October 2008 (373 mm and 367 mm), November 2009 (284 mm and 295 mm), and December 2015 (297 mm and 309 mm) with reported SWF incidents have been

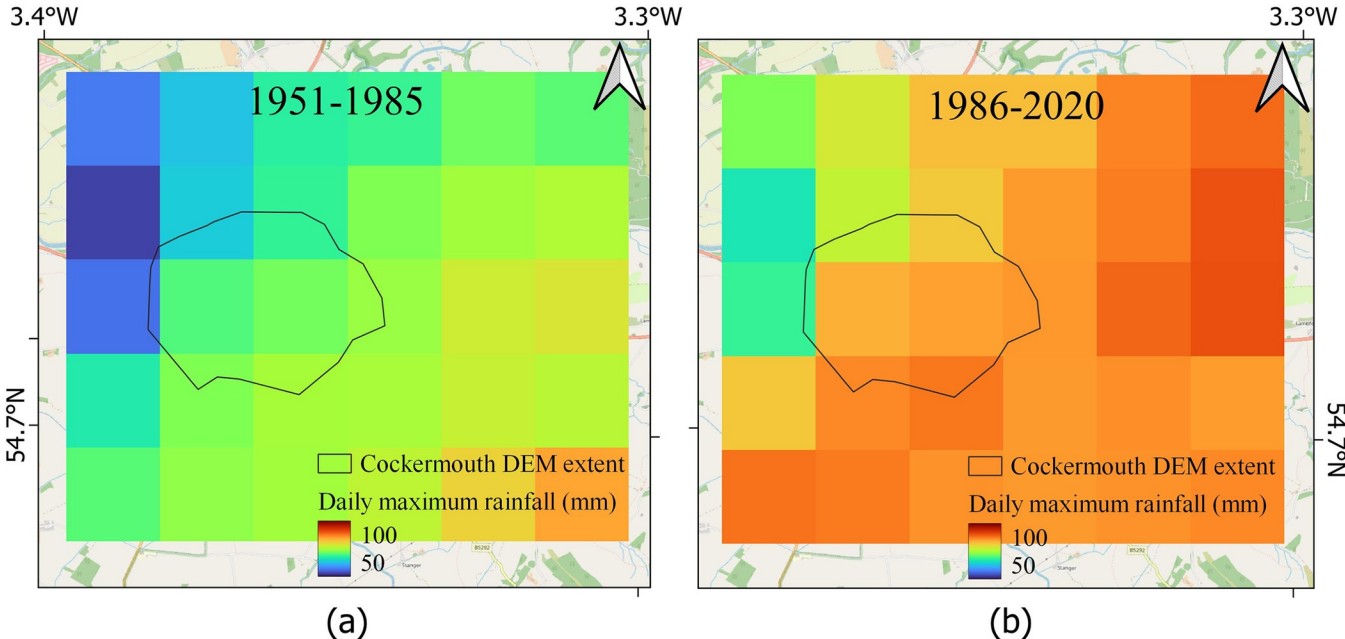

**Fig 6. Estimates of flood hazards for Cockermouth from extreme daily rainfalls.** (a) The amount of maximum daily rainfall in 1951–1985, (b) in 1986–2020.

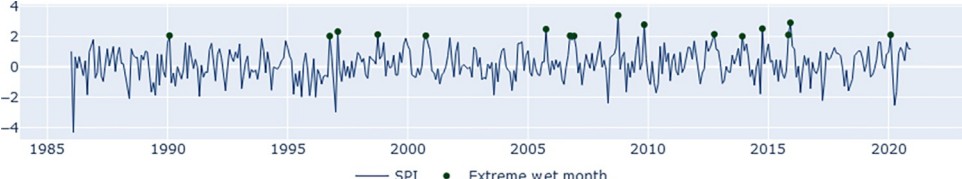

**Fig 7. Monthly SPI series for Cockermouth from 1986 to 2020 and extreme wet months (SPI>2) represented as green dots.**

identified as extreme wet months by HadUK data. However, January 2005 has not been identified as one of the extreme wet months (~150 mm rainfall), though 261 properties were reported to be flooded in this month [23].

We then compared the monthly HadUK precipitation with that of GPM (Fig 8). The time series plot of monthly precipitation indicates that the volatility and general patterns are similar for the two time series. However, the absolute values of rainfall are often larger in HadUK data than GPM data, especially for lower values of precipitation. The Pearson correlation coefficient at 0 to 11 months lags produced maximum correlation of 0.72 (p = 0) at lag 0.

We studied the 5th December 2015 flood event of Cockermouth to understand the closeness of daily total rainfall values to observed rainfall and plotted the rainfall from 1st to 10th December 2015 using HadUK and GPM data (Fig 9). We observed that the GPM data correctly captured an extreme rainfall event on 5th December, whereas the value was much lower in HadUK data. Both HadUK and GPM data indicate a prolonged rainfall event from 3rd to 5th December.

The above results indicate that there is a tendency in HadUK model to overestimate the lower values of rainfall and underestimate the extreme values of rainfall, thus smoothing the overall pattern. This is perhaps expected as HadUK data is based on mathematical interpolations of the rainfall data available from nearby stations. Unavailability of meteorological stations near Cockermouth may have resulted in smoothed values by interpolations from distant stations. In addition, GPM data represents a grid of 10 km resolution. SWF incidents may be caused by local storms of much smaller area, and therefore the 123 mm rainfall on December 5, 2015, captured by GPM data may be a reason of averaging over a larger grid area. Nonetheless, it is evident that remote sensing precipitation observations capture the reality much more accurately than rainfalls estimated by interpolation of station observations based on HadUK data. However, GPM data is not available before 2000 and therefore for long-term understanding of rainfall for any location in England, HadUK is the best dataset available to us. Though the magnitude of rainfall on December 3rd to 5th 2015 is only around 40 mm based on HadUK data, but December 3rd to 5th have been identified as days with extreme rainfall events, which is the focus of our analysis.

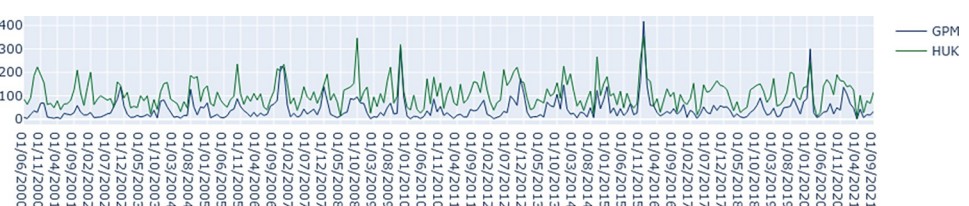

**Fig 8. Time series of monthly precipitation of HadUK (HUK) and GPM data for Cockermouth region.** Horizontal axis represents months/years, vertical axis represents the total monthly rainfall in mm.

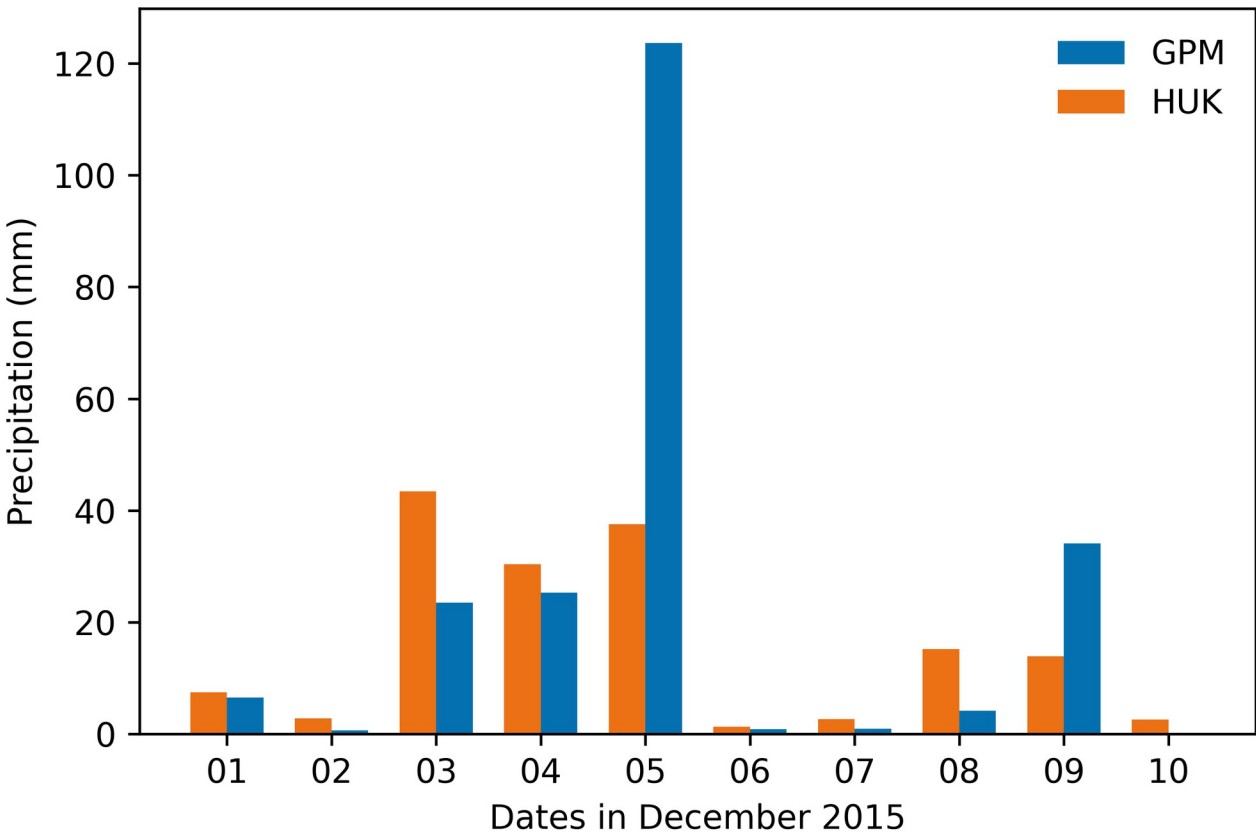

**Fig 9. Comparison of daily precipitation of HadUK and GPM data from 1st December to 10th December 2015, for Cockermouth Region.**

### 4.2 Surface water flood exposure

The TWI values for Cockermouth were within the range -10.7 to 22.4 based on LiDAR data. Based on the photographic DEM, the values were in the range -17.6 to 18.5. Fig 10 shows an example of how TWI based on air photo and LiDAR DEMs represents the local flood risks. Overall, the TWI values were slightly lower on the photographic DEM compared to those based on the LiDAR DEM. The area falling in the open grounds (e.g., garden) side of the image, as well as the area next to the building, roadside, streets, and part of the parking areas show larger TWI values, where chances of water accumulation are expected to be high. The TWI image based on LiDAR DEM at 25 cm resolution seems to capture some distinct paths for water flow and accumulation represented by dark blue lines. For the 10 cm DEM, these lines are not so distinct, but similar locations have been identified to have higher TWI values. Overall, based on visual interpretation, the spatial variations of TWI values look similar on both DEMs.

From the flood resilience and resistance information, we observed that 29% (108) of the total number of residential buildings (365) which were flooded do not have flood resistance features installed and therefore categorised as high exposure (category 4), and almost 23% (83) had measures offering $\geq$ 100 cm of flood protection, and therefore classified as very low exposure (category 1). The remaining buildings were in intermediate risk categories 2 and 3 depending on the height of the PFR feature installed (Table 1).

Table 2 shows the number of buildings in different exposure classes based on EM, EH and a combination of both EM (based on LiDAR and photographic DEM) and EH. Using the TWI data and percentile thresholds, 92 buildings out of 365 were categorised as having very low (1)

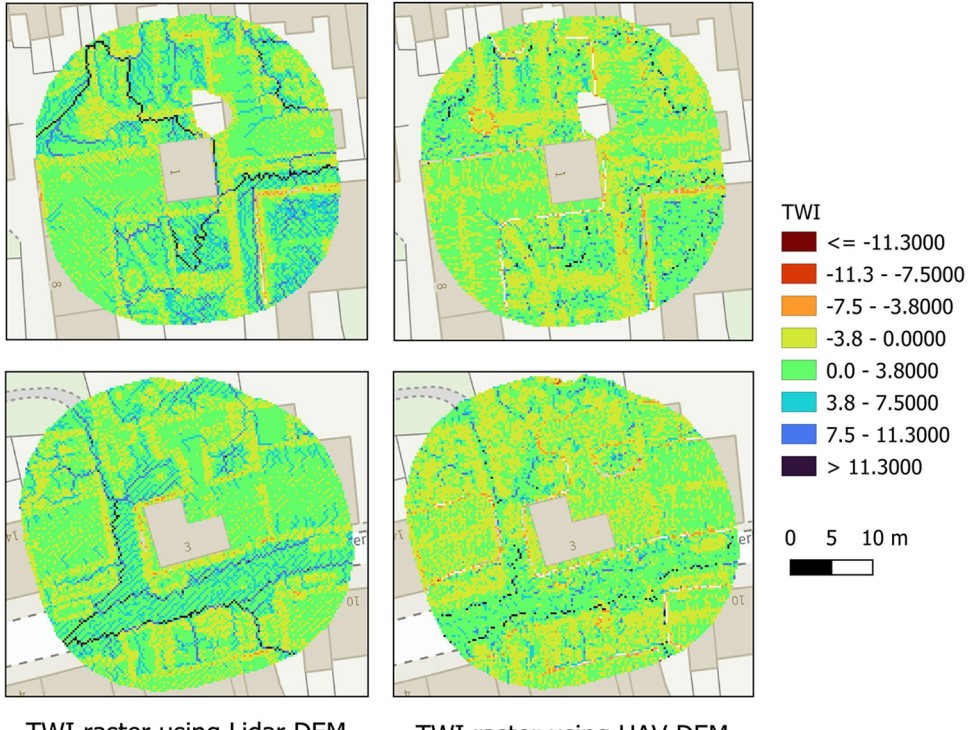

**Fig 10. TWI for the pixels within the buffered polygon surrounding a building extracted from LiDAR and photographic DEM.** Background is from OS Maps © Ordnance Survey 2024 (https://osdatahub.os.uk/dashboard).

exposure and 91 buildings in each of the three higher exposure classes (2 to 4). We added the exposure categories of each building available from TWI and PFR data to prepare the final exposure categories as described in section 3.2.3. We observe that the number of buildings in different exposure categories changes after the application of PFR, and most of the buildings move to very low exposure category because of having PFR feature installed (Table 2). However, there are still a few buildings (13 based on LiDAR based exposure, 20 based on airphoto DEM based exposure) at high exposure from SWF. This could be due to microtopography structure surrounding the buildings that need better PFR to avoid damage from SWF as observed from Table 2.

## 4.3 Effect of DEM resolution to capture SWF exposure microtopography

To understand how the resolution of the DEMs are capturing the risks from microtopography differently, we built a confusion matrix between the risk categories of different buildings

**Table 2. Number of buildings in different exposure classes (EM: Exposure based on microtopography; EH: Exposure based on PFR height) based on microtopography and PFR height.**

| Exposure class | Number of buildings based on microtopography (EM) | Number of buildings based on PFR (EH) | Number of buildings based on EM (LiDAR) and EH | Number of buildings based on EM (airphoto DEM) and EH |
|---|---|---|---|---|
| Very low | 92 | 83 | 229 | 218 |
| Low | 91 | 43 | 67 | 77 |
| Medium | 91 | 131 | 56 | 50 |
| High | 91 | 108 | 13 | 20 |

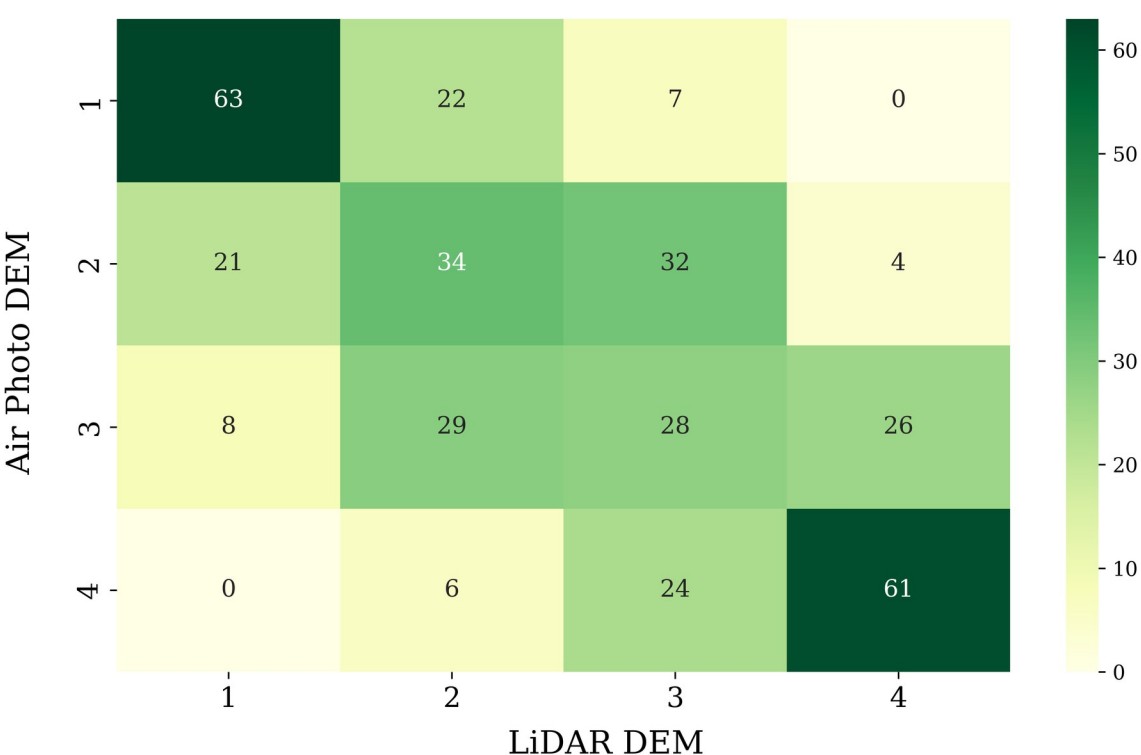

**Fig 11. Confusion matrix to evaluate the closeness of flood exposure for different buildings using LiDAR and air photo DEMs.**

identified by the two different DEMs. 61 of the 91 buildings were identified to be at the highest risk (category 4) using both DEMs. There were some differences in the allocation of buildings at intermediate exposure (categories 2 and 3) using the two DEMs (Fig 11).

## 5. Discussions

We developed a framework to estimate how the SWF hazard from rainfall can vary both spatially and temporally. We used a specific study area (Cockermouth) and made several assumptions to decide whether the flood hazard has increased in the region and how exposed are the individual residential properties to SWF.

Our first assumption is the chosen periods of rainfall for assessing temporal changes. In general, at least 20–30 years of climate data is required to estimate the average climate of any region [35]. Therefore, we used the first 35 years of HadUK data to understand the average/ standard rainfall patterns and used the other half of the period for identifying the deviations.

Our results indicate that the buildings can be classified in different exposure categories based on the surrounding microtopography available from high resolution DEMs and PFR databases. Our proposed approach provides a way to identify those buildings that would benefit from PFR measures and thus could be used to reduce current and future SWF risks. We assumed that a 100 cm and above PFR height provides complete protection of a building from SWF events based on literature survey. This may change with time if the amount and intensity of rainfall increases and/or drainage capacity reduces, thus increasing the flood depth. The height can easily be updated in the framework.

We assumed that a 15 m buffer representing the neighbourhood determines the SWF exposure of each building. We consider 15 m to be a realistic assumption, but this can again be adjusted based on availability of suitable ground truth data. We assumed median of the raster

TWI values surrounding each building to be a representative exposure from microtopography. We chose median over mean as it is more robust to outliers and a better representative of average for a skewed distribution. However, we did not consider the flow patterns in the TWI image, which could also indicate the drainage routes, nor the efficiency of existing drains, as this data is not available with us. Based on visual interpretation, there are some clear paths of water flow on the LiDAR DEM based TWI image (25 cm), which are not easily distinguishable from the UAV based TWI (10 cm), although the latter is at a higher resolution. Such differences may arise because of differences in horizontal and vertical accuracies of the DEMs, as well as the DEM resolutions [48]. The topographic features with smaller scales than the resolution of a DEM are not captured by the DEM. Therefore, the 10 cm DEM captured more subtle details in the variation of topography. Some authors have studied the effects of DEM resolution to capture topographic variations and observed an improvement in hydrological simulations with increasing resolutions of DEMs [43,49,50] and concluded that finer resolution DEMs capture the details much better than coarser resolution DEMs. However, the resolutions they compared were higher than 1 m. Thus, determining the optimal DEM resolution for identifying SWF exposure using sub meter resolution DEMs and improving SWF modelling remains a topic of future research. Our results suggest that TWI based on high resolution DEMs collected using both aircraft laser and photographic sensors mounted on UAVs can be used as a metric to identify the risks for individual properties from surrounding microtopography based flood exposure. As both DEMs provide horizontal and vertical accuracies better than ±10 cm, they can capture the microtopography details very well (Ramachandran et al. 2023 [1]). However, UAVs present the added advantage of on-demand, autonomous and cost-effective deployment whereas aircraft-LiDAR solutions can provide wide area coverage but at a higher operational cost [51].

## 5.1 Current status of SWF management practice in England and usefulness of the proposed method

Our proposed methodology to estimate flood hazard aligns with the UK Government's plans to develop a comprehensive set of indicators [52] to estimate environmental changes. The proposed flood hazard visualization based on some metrics would be a useful measure to rapidly estimate how the risks from flood hazard varies both spatially (across England) and temporally (past and future) and could be used to improve the resilience of localities from flood hazard, from both fluvial and SWF. In England, around 5 million properties are at risk from SWF and SWF is the most widespread flood risk in England at present [10]. The government has recently set out their SWF management action plan [10], which involves better risk assessment, better SWF mapping and more resilient infrastructure, among other measures. 28 lead local authorities were supported by the Government in 2021 to carry out further studies to improve the detailed information on local SWF risk for 3.3 million people [53]. As SWF is a localised phenomenon, information about local rainfall, topography, drainage networks and PFR are required to better model and manage SWF. Some of this knowledge is available within the Lead Local Flood Authorities (LLFA), insurance agencies and water and sewerage companies [15]. From 2020, the Environment Agency encouraged different organisations to share the data and information available to improve SWF mapping for different localities in England [53]. Such initiatives helped improve local SWF mapping for around 3,500 km$^2$ across England which covers around 3 million residential properties [53]. The proposed method used here could be implemented to identify the locations in England where SWF hazard has increased in recent years (HadUK data). UAVs can be used to capture stereo images of those regions, generate elevation models and microtopographic information to find the properties at higher risk

from SWF that could be protected by installing PFRs. Such information would also assist the insurance industry in setting premiums for a property, depending on the SWF exposure of each building. SWF forecasting is still quite challenging due to the uncertainties in the severity of the local convective rainfall [10]. However, if properties vulnerable to SWF could be identified and protected by installing PFR features, damage from SWF could be significantly reduced.

For Cockermouth and surrounding areas, the flood hazard estimation based on HadUK data suggest that the extreme rainfall events are increasing in this area. This increase can be correlated with climate change and increased temperature, because of which there is an increase of water holding capacity in the atmosphere, leading to more rainfall [54]. Increased rainfall events reduce soil moisture deficit that reduces the drainage capacity, thus increasing the chances of SWF events. Such incidence was observed in Bedford in December 2020 as it was the second wettest December from 1981 since the records began [55]. For better management of damages from SWF hazards, it is therefore important that such locations are identified and the properties in such locations be protected by installing PFR features.

## 5.2 SWF hazard in the coming decades

We have already witnessed an increase in SWF in the recent decades with an increase in population and an expansion in urban areas. Both are expected to continue in the coming decades. A combination of increased rainfall leading to decreasing soil moisture deficit, and urban area expansion could translate into a larger number of properties being affected by SWF. It is possible to use the proposed framework to estimate the flood risk/hazard with climate for other regions in England and identify the regions where the flood hazards have increased most in the recent decades. However, as we observed that the modelled rainfall estimates most often do not capture the actual rainfall amounts, therefore, to analyse the real extents of hazards from flood incidents we will need data from other sources. Remotely sensed precipitation measurements, such as GPM could be an alternative, but this is available at 10 km resolution and from 2000, thus not enough for understanding the long-term changes in SWF events. The rain gauge stations provide the most accurate estimates of rainfall and therefore most useful for flood modelling. Our approach could identify the most vulnerable locations to focus (hazard visualisation) for flood modelling and the properties in those locations most vulnerable to SWF.

## 6. Conclusions

In this study, we proposed a framework for identifying the most vulnerable locations to flash flooding by analysing long-term daily and monthly rainfall data obtained from past records. It was observed that hazard from rainfall has increased in Cockermouth in the recent period. The same methodology can be applied on a country scale to identify regions suffering from SWF hazards. The minimum monthly rainfall leading to flood hazards varies spatially and we need long-term rainfall data for any region to identify/forecast the minimum rainfall amount that could lead to flood hazards. Subsequently, high-resolution DEMs can be used to define property level exposure using micro-topography based index, such as TWI. Properties at different levels of risks to flash flooding can be identified. Our analysis, using both 10 cm and 25 cm DEMs, shows that the characterization of exposure is similar between the two models. A future study could focus on determining the optimal DEM resolution for identifying SWF exposure and SWF modelling.

Our study also revealed that many properties without PFRs are highly exposed to flash flooding and should be protected to avoid damage in the future. By combining property-level

information from the DEMs with postcode-based SWF extent, depth, and velocity data provided by the UK Government, property owners and the insurance industry can better estimate the subtle variations of SWF exposure at the property level and make informed decisions about necessary flood protection measures.

## Acknowledgments

Authors are thankful to the reviewers for their constructive comments, which have improved the quality of the manuscript. We also extend our gratitude to the climate data providers at the Hadley Climate Centre, UK for providing the HadUK data, and to the National Aeronautics and Space Administration (NASA) for supplying the GPM data.

## Author Contributions

**Conceptualization:** Kriti Mukherjee, Mónica Rivas Casado.

**Data curation:** Mónica Rivas Casado, Rakhee Ramachandran.

**Formal analysis:** Kriti Mukherjee.

**Funding acquisition:** Mónica Rivas Casado.

**Investigation:** Kriti Mukherjee.

**Methodology:** Kriti Mukherjee.

**Project administration:** Mónica Rivas Casado.

**Resources:** Kriti Mukherjee, Mónica Rivas Casado, Rakhee Ramachandran.

**Software:** Kriti Mukherjee.

**Supervision:** Mónica Rivas Casado.

**Validation:** Kriti Mukherjee.

**Visualization:** Kriti Mukherjee.

**Writing – original draft:** Kriti Mukherjee.

**Writing – review & editing:** Kriti Mukherjee, Mónica Rivas Casado, Paul Leinster.

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
