## [Decision Letter · Decision Letter 0]

10 Jun 2024

PONE-D-23-36834Harnessing long-term gridded rainfall data and microtopographic insights to characterise risk from surface water floodingPLOS ONE

Dear Dr. Mukherjee,

Thank you for submitting your manuscript to PLOS ONE. After careful consideration, we feel that it has merit but does not fully meet PLOS ONE’s publication criteria as it currently stands. Therefore, we invite you to submit a revised version of the manuscript that addresses the points raised during the review process. As the editor I took the liberty in looking at your manuscript after receiving the first reviewer's feedback because we have been waiting too long for the second reviewer to get back to us and your manuscript topic is in my field of expertise. I agree with all the comments received and have no additional points to add. In your response to the comments and revised version, please ensure you address all comments.

We look forward to receiving your revised manuscript.

Kind regards,

Guy J-P. Schumann

Section Editor

PLOS ONE

3. We note that Figures 1 and 13 in your submission contain [map/satellite] images which may be copyrighted. All PLOS content is published under the Creative Commons Attribution License (CC BY 4.0), which means that the manuscript, images, and Supporting Information files will be freely available online, and any third party is permitted to access, download, copy, distribute, and use these materials in any way, even commercially, with proper attribution. For these reasons, we cannot publish previously copyrighted maps or satellite images created using proprietary data, such as Google software (Google Maps, Street View, and Earth). For more information, see our copyright guidelines: http://journals.plos.org/plosone/s/licenses-and-copyright.

1. You may seek permission from the original copyright holder of Figures 1 and 13 to publish the content specifically under the CC BY 4.0 license.  

Additional Editor Comments (if provided):

Reviewers' comments:

Reviewer's Responses to Questions

**Comments to the Author**

1. Is the manuscript technically sound, and do the data support the conclusions?

Reviewer #1: Yes

2. Has the statistical analysis been performed appropriately and rigorously? 

Reviewer #1: Yes

3. Have the authors made all data underlying the findings in their manuscript fully available?

Reviewer #1: No

4. Is the manuscript presented in an intelligible fashion and written in standard English?

Reviewer #1: Yes

5. Review Comments to the Author

Reviewer #1: The authors have done wonderful study by understanding the micro topography to estimate the surface water flooding vulnerability at Cockeremouth in UK. Such study will be very useful to plan mitigation measures in vulnerable areas in other places of the world.

However, there are certain observation the authors needs to elaborate upon

(i) What is the extent of accuracy of rainfall data obtained fom HadUK. The authors have mentioned that they have done pearson coeffcient analysis with GPM data but have not adequately described in the text.

If pearson test is conducted then the authors may present the relationship between GPM and HadUK data

(ii) The authors have mentioned regarding the depth of flood that is hazardous but they could have categoriesed minimum rainfall amount that can be considered as threshold.

(iii) is objective 2 needed at this point of time.

(iv) Whether monthly rainfall amount can identify the rainfall hazard

(v) The authors should have referred to some other already published studies about reliability of gridded precipitation data with ground based observed data on different temporal scale such as daily and monthly.

The authors have compared the HadUK data with GPM data in this present study. GPM data might not give accurate rainfall amounts in given area.

(vi) Projection data is for future scenarios, so I think the authors could have concentraterd more on application of methodology and standardized it.

(vii) The authors have mentioned about the TWI, but what is the scenario of spilling of both the rivers as it is at the confluence point.

6. PLOS authors have the option to publish the peer review history of their article (what does this mean?). If published, this will include your full peer review and any attached files.

Reviewer #1: No

---

## [Author Response · Author response to Decision Letter 0]

12 Aug 2024

The authors are thankful to the editor and the anonymous reviewer for their constructive comments on the article. Below we provide answers to the queries and concerns. 

• We ensure that our manuscript meets PLOS ONE’s style requirements.

2. We note that you have indicated that there are restrictions to data sharing for this study. PLOS only allows data to be available upon request if there are legal or ethical restrictions on sharing data publicly. 

a. If there are ethical or legal restrictions on sharing a de-identified data set, please explain them in detail (e.g., data contain potentially identifying or sensitive patient information, data are owned by a third-party organization, etc.) and who has imposed them (e.g., a Research Ethics Committee or Institutional Review Board, etc.). Please also provide contact information for a data access committee, ethics committee, or other institutional body to which data requests may be sent.

• We updated the data availability statement in our revised manuscript. Data can be available from a repository on request and approval.

b. If there are no restrictions, please upload the minimal anonymized data set necessary to replicate your study findings to a stable, public repository and provide us with the relevant URLs, DOIs, or accession numbers. For a list of recommended repositories, please see

• As above

4. We note that Figures 1 and 13 in your submission contain [map/satellite] images which may be copyrighted. All PLOS content is published under the Creative Commons Attribution License (CC BY 4.0), which means that the manuscript, images, and Supporting Information files will be freely available online, and any third party is permitted to access, download, copy, distribute, and use these materials in any way, even commercially, with proper attribution. For these reasons, we cannot publish previously copyrighted maps or satellite images created using proprietary data, such as Google software (Google Maps, Street View, and Earth). For more information, see our copyright guidelines: http://journals.plos.org/plosone/s/licenses-and-copyright. 

• We replaced Figures 1 and 13 (Figure 10 in the revised manuscript) by replacing the basemap by Ordnance Survey maps available in the public domain and provided appropriate referencing as suggested on their website (https://osdatahub.os.uk/, https://www.nationalarchives.gov.uk/doc/open-government-licence/version/3/) 

• The reference style has been modified to comply with the journal requirement. 

Reviewer #1: The authors have done wonderful study by understanding the micro topography to estimate the surface water flooding vulnerability at Cockeremouth in UK. Such study will be very useful to plan mitigation measures in vulnerable areas in other places of the world.

The authors thank the reviewer for the encouraging comments. 

However, there are certain observation the authors needs to elaborate upon

(i) What is the extent of accuracy of rainfall data obtained fom HadUK. The authors have mentioned that they have done pearson coeffcient analysis with GPM data but have not adequately described in the text.

If pearson test is conducted then the authors may present the relationship between GPM and HadUK data

We agree with the reviewer. We forgot to report on the values of Pearson coefficient in the results. We have now included it in the revised manuscript. Please see L404-405.

(ii) The authors have mentioned regarding the depth of flood that is hazardous but they could have categoriesed minimum rainfall amount that can be considered as threshold.

The minimum monthly rainfall amount that could be hazardous for Cockermouth has been identified and mentioned in the manuscript (L391). However, the minimum daily rainfall amount could not be categorized because the short term (daily or hourly) extreme values are not captured well by the interpolated gridded data as we observed for the 2015 December incident (Figure 9). 

(iii) is objective 2 needed at this point of time.

We removed objective 2 and related analysis from the revised manuscript. We have now two objectives in our revised manuscript and eleven figures. Figure 13 of the original manuscript is Figure 10 in the revised manuscript.

(iv) Whether monthly rainfall amount can identify the rainfall hazard

Monthly rainfall amount can identify rainfall hazard if long-term monthly rainfall time series is available. Based on the HadUK data for Cockermouth Region, we observed that the value is approximately 200 mm (mentioned in L391 in the revised manuscript). The same methodology can be applied to quality checked station based rainfall data, such as from MIDAS (https://catalogue.ceda.ac.uk/uuid/220a65615218d5c9cc9e4785a3234bd0) and would be more accurate. There is no station data available for Cockermouth Region, but HadUK data used in this study was generated using MIDAS data (Hollis et al., 2019). 

(v) The authors should have referred to some other already published studies about reliability of gridded precipitation data with ground based observed data on different temporal scale such as daily and monthly.

The authors have compared the HadUK data with GPM data in this present study. GPM data might not give accurate rainfall amounts in given area.

We agree that GPM data may not give accurate rainfall amounts because of its coarser resolution (10 km). However, as there are no historical station observations available for Cockermouth Region, therefore, GPM data was one of the options available to us for validation. We also added a few lines in the revised manuscript that HadUK data went through several quality controls before it was published in the public domain (L125-126, L251-257).

(vi) Projection data is for future scenarios, so I think the authors could have concentraterd more on application of methodology and standardized it.

We removed the future scenarios and related analysis from the revised manuscript.

(vii) The authors have mentioned about the TWI, but what is the scenario of spilling of both the rivers as it is at the confluence point.

Spilling of both rivers would be considered a result of fluvial flooding. As we are only focusing this paper on surface water (pluvial) flooding, we have not considered this factor into account.

---

## [Editor Report · Decision Letter 1]

6 Sep 2024

Harnessing long-term gridded rainfall data and microtopographic insights to characterise risk from surface water flooding

PONE-D-23-36834R1

Dear Dr. Mukherjee,

We’re pleased to inform you that your manuscript has been judged scientifically suitable for publication and will be formally accepted for publication once it meets all outstanding technical requirements.

Kind regards,

Guy J-P. Schumann

Section Editor

PLOS ONE
---

## [Editor Report · Acceptance letter]

13 Sep 2024

PONE-D-23-36834R1 

PLOS ONE

Dear Dr. Mukherjee, 

I'm pleased to inform you that your manuscript has been deemed suitable for publication in PLOS ONE. Congratulations! Your manuscript is now being handed over to our production team.

Kind regards, 

on behalf of

Dr. Guy J-P. Schumann 

Section Editor

PLOS ONE